# Overexpression of *Medicago sativa LEA*4-4 can improve the salt, drought, and oxidation resistance of transgenic *Arabidopsis*

Huili Jia[1,2], Xuemin Wang[3], Yonghong Shi[2], Xinming Wu[2], Yunqi Wang[2], Jianning Liu[2], Zhihong Fang[2], Chunyan Li[2], Kuanhu Dong[1]*

**1** College of Animal Science and Technology, Shanxi Agricultural University, Taigu, Shanxi, China, **2** Animal Husbandry and Veterinary Institute, Shanxi Academy of Agricultural Sciences, Taiyuan, Shanxi, China, **3** Chinese Academy of Agricultural Sciences, Institute of Animal Science, Beijing, China

* dongkuanhu@126.com

**Data Availability Statement:** All relevant data are within the manuscript and its Supporting Information files.

## Abstract

Late embryogenesis abundant (LEA) proteins are widely involved in many adverse conditions among plants. In this study, we isolated a *LEA*4 gene from alfalfa (*Medicago sativa* L.) termed *MsLEA*4-4 via a homology cloning strategy. *MsLEA*4-4 encodes 166 amino acids, and the structural analysis showed that the gene contained five repeating TAQAAKEKTQQ amino acid motifs. There were a large number of α-helix in *MsLEA*4-4, and belongs to hydrophilic amino acid. Subcellular localization analysis showed that *MsLEA*4-4 was localized in the nucleus. The *MsLEA*4-4 promoter consisted of G-box and A-box elements, abscisic acid-responsive elements (ABREs), photo regulation and photoperiodic-controlling *cis*-acting elements, and endosperm expression motifs. The *MsLEA*4-4 overexpressing in *Arabidopsis* conferred late-germination phenotypes. Resistance of the overexpressed plants to abiotic stress significantly outperformed the wild-type (WT) plants. Under salt stress and abscisic acid treatment, with more lateral roots and higher chlorophyll content, the overexpressed plants has a higher survival rate measured against WT. Compared to those in the WT plants, the levels of soluble sugar and the activity of various antioxidant enzymes were elevated in the overexpressed plants, whereas the levels of proline and malondialdehyde were significantly reduced. The expression levels of several genes such as *ABF*3, *ABI*5, *NCED*5, and *NCED*9 increased markedly in the overexpressed plants compared to the WT under osmotic stress.

## Introduction

During growth and development, plants tend to be subjected to various abiotic stresses, such as drought and flood disasters, extreme temperatures, and soil salinity, which can lead to hydropenia due to their immobility [1–3]. Dehydration can result in a series of physiological disorders in plants, such as respiration and photosynthesis disorders [4–6]. In the long process of evolutionary, plants have formed their own protection mechanisms to withstand the effect

**Funding:** The work was supported by a grant from the Special Funds for the Technical System of the Modern Agricultural Industry of China (CARS-34), by the Species Resources Conservation of China (2130135) and by the Key Research and Development Projects of ShanXi (201703D211002-9-2).

**Competing interests:** The authors have declared that no competing interests exist.

of drought, salinity and low temperature, to ensure their normal growth and development [7–9]. Various transcription factors, antioxidant enzyme genes and functional proteins play important roles in this process [10]. LEA, or dehydrin among the proteins involved in the mechanism of various abiotic, features prominently in plant abiotic stress resistance and cell repair and can be activate by calcium signals [11].

LEA proteins were first found to accumulate at the late stage of cotton (*Gossypium* spp.) seeds development [12,13]. They can also be found in vegetative tissues in plants subjected to abiotic stress, and in bacteria and some invertebrates [14, 15]. LEA proteins are key proteins involved in cell antistress protection that can prevent protein aggregation caused by drought and improve plant tolerance to drought [16,17]. LEA protein synthesis can protect biological membranes from damage at low temperature and increase the resistance of plants to freezing conditions [18,19]. These proteins can also reduce osmotic pressure by stabilizing cell membranes, isolating or binding ions, and reducing cell dehydration and structural damage under hyperosmotic and hypersaline conditions [20]. Therefore, their expression is bound up with the antistress protection of plant cells [21]. With the development of modern biological technologies, LEA proteins in various species have been identified and proved to be characterized by resistance to drought, cold and salt [22]. The *LEA* gene family was first reported in *Arabidopsis thaliana*. There were 34, 35, 78, and 32 LEA proteins have been detected in rice (*Oryza sativa*) [23], cucumber (*Cucumis sativus*) [24], and corn (*Zea mays*) [25]. In *Medicago truncatula*, 23 LEA proteins have been found [26]. Upland cotton contains a variety of *LEA* genes; there are 242, 136 and 142 LEA in *Gossypium hirsutum*, *Gossypium arboretum* and *Gossypium raimondii* [27].

Various LEA proteins exhibit high hydrophilicity and thermal stability, and are rich in glycine or other amino acids, such as alanine, serine, and threonine [23]. Studies have focused more on LEA 1, LEA 2, and LEA 3 than on other LEA proteins. LEA 1, with the ability to improve plant stress tolerance, has been widely studied in soybean [28,29]. LEA 2 and LEA 3 have been investigated in various plants, but further studies on wheat should be performed [30]. LEA 4 also ubiquitous in plants, had been cloned from *Arabidopsis*, corn, soybean, and caragana (*Caragana korshinskii*), and it had been shown to be most abundant in the *LEA* family of *A. thaliana* [31,32]. Due to a large number of a-helix in LEA4, most of them have a strong hydrophilicity, which favors cells to enhance the water absorption under drought stress [33]. *LEA*4 has been transferred into *A. thaliana*, *Oryza sativa* and other plants, and the overexpressed plants show high resistance to salt and drought stress [34]. Several studies, such as expression pattern, cellular localization, transgenic plant-related, and protein characterization studies, have obtained data on the protective functions of LEA1-3. However, studies on LEA4 are relatively scarce, and the function and mechanism of action of this protein remain unclear.

Alfalfa (*Medicago sativa* L.), a perennial legume rich in crude protein, vitamins, and various minerals, has been widely cultivated across the world. Ruminant such as cows and goats demonstrate a preference for this legume with high dietary fiber [35,36]. This crop is digestible and well-absorbed, and is primarily used as feed in the form of dried hay, haylage, and grazing material dairy cow, sheep, and beef production systems [37]. Alfalfa has a long resistance to drought stress during its growth, and its yield does not decrease under 50 mM salt stress [38]. However, adverse external conditions, such as high salinity, drought, high temperature, and other types of stress, negatively affect the yield and quality of alfalfa [39,40]. Therefore, breeding alfalfa varieties resistant to salt, drought, and low temperature represent the goal and task of modern breeders [41]. Several stress-related genes, such as those encoding zinc finger proteins and transcription factors and those participating in metabolic pathways, have been cloned from alfalfa [42]. Analyzing the mechanisms of action of genes related to abiotic stress and providing a basis for breeding resistant alfalfa are the emphasis and challenges of current studies [43,44].

In this study, we isolated and characterized a *LEA*4 gene from *M. sativa* designated *MsLEA*4-4. The expression of this gene was influenced by abiotic stresses, such as salt, drought, and low temperature. *MsLEA*4-4 overexpressing in *A. thaliana* improved the antioxidant capacity of the resulting transgenic plants and significantly increased stress resistance.

## Materials and methods

### Plant materials

The Zhongmu No. 1 strain from the Beijing Animal Husbandry and Veterinary Research Institute of the Chinese Academy of Agricultural Sciences was utilized in this study. The seeds were disinfected with chlorine and germinated at 24˚C. Then the seedlings were transferred to nutrient solution to grow until the time the first true leaf emerged, and the grow condition was 26˚C/22˚C, 16 h light/8 h dark.

In this study, the Columbia-0 *A. thaliana* was utilized. The seeds were planted on 1/2 MS nutrient medium after sterilized with ethanol and sodium hypochlorite and washed with double-distilled water. The seedlings were transferred to soil to grow after 4 weeks amid 22˚C/18˚C, 16 h light/8 h dark.

### Stress treatments

The abiotic stresses in this experiment were NaCl (0, 50, 100, 150, 200, 250, and 300 mM), $H_2O_2$ (200 mM), NO (0.1 mM $Na_2Fe(NO)_5$), drought (100, 150, and 200 mM mannitol), and low temperature (4˚C). The plants were also subjected to light stress (0, 6, 12, 24, 48, and 72 h), dark stress (0, 6, 12, 24, 48, and 72 h), light recovery after dark stress (15 min, 30 min, 2, 12, 24, 48, and 72 h), and photoperiod stress (0 h light/24 h dark, 4 h light/20 h dark, 8 h light/16 h dark, 12 h light/12 h dark, 16 h light/8 h dark, 20 h light/4 h dark, and 24 h light/0 h dark).

### Phytohormone treatments

The phytohormone treatments in this experiment were abscisic acid (ABA) (0.1 mM) and gibberellins (GA) (100 mg/ml). The plant growth condition was identified in ABA and GA treatment.

### Vector construction and transformation

According to the unigene of similar *LEA* sequences obtained from transcriptome sequencing, *LEA*4-4 from *M. sativa* L. was cloned using the rapid amplification of cDNA ends (RACE) technique with the primers 5′–AAAGAATGGCATCCCACGACC–3′ and 5′–CTCAATCAA CAACGTTACGACGG–3′. A 498 bp *MsLEA*4-4 opening reading frame (ORF) was amplified. The ORF sequence containing the restriction enzyme *Bgl*II and *Bste*II (S1 Table) recognition sites were inserted into the pCAMBIA3301 vector (S1A Fig). GV3101 strain was served to transform the *MsLEA*4-4 ORF to *A. thaliana* by inflorescence infection. The overexpressed plants were filtered on 1/2 MS nutrient medium containing glufosinate-ammonium (50 mg/L) and analyzed by PCR and quantitative real-time PCR (qRT-PCR) on the genomic and transcriptional levels, respectively. The screening procedure was repeated until T3 transgenic plants were obtained; these plants were used in subsequent experiments.

### Subcellular localization

The *MsLEA*4-4 ORF sequence, which contained *Xhol* and *Spe*I recognition sites (S1 Table), was inserted into the pA7-green fluorescent protein (GFP) vector (S1B Fig). *MsLEA*4-4-pA7-GFP fusion expression vector was imported into onion epidermal cells with a gun

bombardment, and the GFP gene were expressed instantaneously. Then, the fluorescent signal was observed after incubation for 20 h in darkness at 25˚C.

## Southern blot analysis

The *MsLEA*4-4 coding sequence was labeled with digoxigenin (DIG) using DIG probe synthesis kit (Roche) and used as a hybridization probe. DNA was extracted from 2-week-old overexpressed plants and digested 16 h with restriction enzymes *EcoR*I and *Hind*III. The prepared DNA was run on a 0.7% agarose gel electrophoresis overnight under 4˚C and 25 V constant voltages, and then was transferred to a Hybond N$^+$ nylon membrane (Amersham) after 20 hours. The membrane was marked, washed with 2×SSC (salinesodium citrate buffer), baked at 80˚C for 2 h to fix the DNA, afterwards, the membrane was placed in a hybridization tube containing the hybridization buffer and probe for overnight hybridization at 42˚C. After that, the membrane was respectively washed with 2 × SSC and 1 × SSC for 15 min, then added antibody solution for 30 min incubation. Then the membrane was washed twice for 15 min with washing buffer and equilibrated 5 min in detection buffer. Finally, the membrane was incubated in prepared color substrate solution for 1–3 h in the dark to detect the bands.

## qRT-PCR analysis

Total RNA was extracted from seedlings aged 4 weeks by Trizol (Life Technologies, USA) method, and DNA contamination was removed by DNase. Reverse transcribed total RNA into cDNA according to the RevertAid$^{TM}$ First Strand cDNA Synthesis kit. SYBR Green PCR Master Mix (Takara, Japan) was utilized to qRT-PCR, and the reaction system was 20 μl and need 5 ng cDNA. ABI 7500 was used and two-step method for qRT-PCR was as follows: step 1, 95˚C for 30 s; step 2, 41 cycles of 95˚C for 5 s and 60˚C for 34 s. All primers are listed in S1 Table. The reference genes were *AtActin* and *MsActin2* (S1 Table). The $2^{-\Delta\Delta Ct}$ method acted to measure the expression levels of different treatments according to the obtained Ct values [45]. All experiments have three biological replicates and three technical replicates.

## Growth index measurement

Seeds of T$_3$ overexpressed plants and wild-type (WT) plants were sterilized and seeded on 1/2 MS nutrient medium to germinate for 5 days. After that, the seedlings were transferred to 1/2 MS containing 50 mM NaCl and 150 mM mannitol to grow 7 days. Lateral root numbers was assessed and the fresh weight was measured. Each measurement had 20 seedlings and replicated three times.

The overexpressed and WT seeds were both seeded on 1/2 MS nutrient medium containing 200 mM NaCl, 0.1 mM ABA and 100 mM mannitol. After 72 h of vernalization, the seedlings were grown in an illumination incubator (22˚C/18˚C, 16 h light/8 h dark) for 10 days for the analysis of the germination rate. Each sample comprised 100 seeds and performed on three biological replicates.

For measurement of chlorophyll content, seedlings were grown in an illumination incubator (22˚C/18˚C, 16 h light/8 h dark) for 7 days and transferred to 1/2 MS nutrient medium containing ABA (0.1 mM), $H_2O_2$ (10 mM) and NaCl (300 mM) for another 7-day growth. Chlorophyll was measured according to the methods of Zhang et al. [46]; 0.1 g of seedlings were dipped in 10 ml of 80% acetone and kept away from light for 72 h, and then the absorbance values at wavelengths of 663 and 646 nm were measured. Each adversity stress treatment has three biological replicates.

### Physiological indexes measurements

The content of proline (Pro), malondialdehyde (MDA) and soluble sugar depended on the acid indanone method, thiobarbituric acid method, anthrone colorimetric method, respectively. Activity levels of superoxide dismutase (SOD), peroxidase (POD), and catalase (CAT) activity levels were determined with appropriate kits. All kits were obtained from Nanjing Jiancheng Bioengineering Institute.

### Expression levels of several genes related to osmotic stress

Seeds of $T_3$ overexpressed and WT were seeded on 1/2 MS nutrient medium (22˚C/18˚C, 16 h light/8 h dark) to grow two weeks. Then RNA was extracted from fresh tissue, reverse translation to cDNA for qRT-PCR to detect the expression levels of several genes responsive to osmotic stress (*NCED*9, *NCED*5, *ABF*3, *ABI*5, *RD29A*, *RAB*18, and *P5CS*).

### Statistical analysis

Each experiment in this study was performed with three biological replicates. All data had calculated the mean ± standard deviation (SD), and utilized T-test to analysis. The significance threshold was $P < 0.05$.

## Results

### Cloning and sequence analysis of *MsLEA*4-4 from *M. sativa* L.

According to a 225 bp unigene of similar LEA sequences (unpublished), obtained from transcriptome sequencing, *LEA*4-4 from *M. sativa* L. was cloned via RACE, and several approaches were applied to establish the presence of *LEA*4 gene in alfalfa leaves. The *MsLEA*4-4 fragment contains a 498 bp ORF and encodes a protein of 166 amino acids. The protein was rich in glycine (G), alanine (A), glutamine (Q), lysine (K), arginine (R), and threonine (T), despite a shortage of cysteine (W) and tryptophan (C). In addition, five imperfect 11-mer motifs were presented in the deduced MsLEA4-4 protein (Fig 1A).

The BLAST function in NCBI was used for sequence alignment, and the results showed that *MsLEA*4-4 shared high sequence similarity with LEAs from other species. *MsLEA*4-4 is 96% homologous to the *M. truncatula* sequence and 52%–82% homologous to *A. thaliana*, *Glycine soja*, *Cicer arietinum*, and *Trifolium subterraneum* sequences. Phylogenetic tree analysis showed the similar to *MtLEA*4-4 from *M. truncatula* (Fig 1B). This finding indicated that the gene might take on the role of LEA4-4 protein. Thus, the identified gene was named *MsLEA*4-4.

The MsLEA4-4 amino acid sequence was predicted by Bioinf software (http://bioinf.cs.ucl.ac.uk/psipred/), and showed that there were a large number of α-helices in *MsLEA*4-4 amino acid sequence (Fig 1C). The hydrophilicity/hydrophobicity of *MsLEA*4-4 was predicted by https://web.expasy.org/protscale/. Most of the amino acids in MsLEA4-4 were below 0, the lower the amino acid score, the stronger of the hydrophilicity. Therefore, MsLEA4-4 was composed of hydrophilic amino acids (Fig 1D).

### Expression pattern and subcellular localization of *MsLEA*4-4

Extracted RNAs from the roots, stems, leaves, flowers and seeds of the same alfalfa plant and the expression patterns of *MsLEA*4-4 in alfalfa were analyzed through qRT-PCR. The results showed that *MsLEA*4-4 was expressed in the root, stem, legume, and seeds, with the highest expression in the legume. The relative expression of *MsLEA*4-4 reached 22.65 in the legume, whereas in leaves and flowers it was probably too low to be detected (Fig 2A).

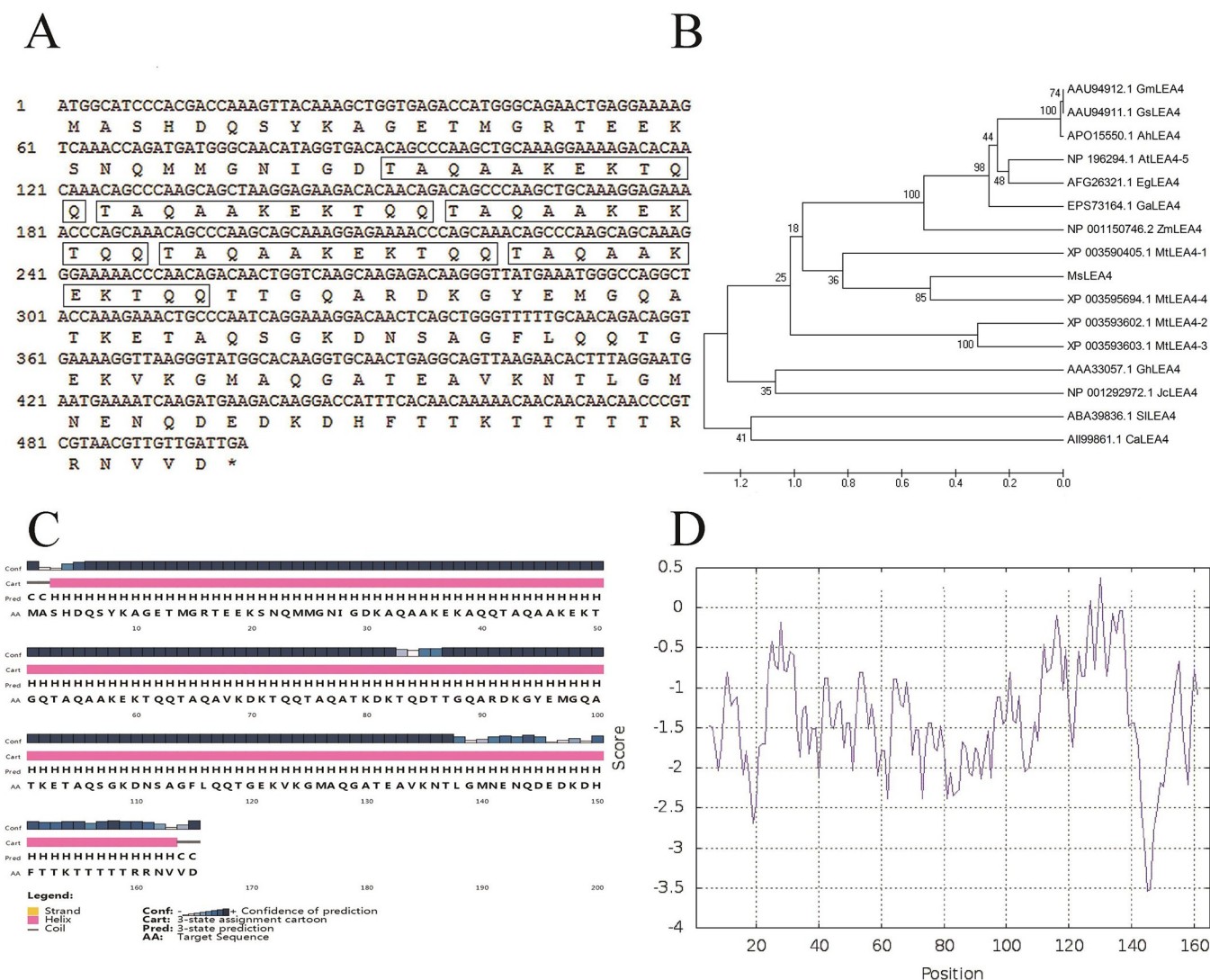

**Fig 1. The sequences and structure analysis of *MsLEA*4-4 from *Medicago sativa*.** (A) ORF sequences and deduced amino acid sequences of *MsLEA*4-4. There are five imperfect 11-mer motifs in the deduced proteins, as shown by the five boxes. (B) Phylogenetic tree analysis of MsLEA4-4 and other LEA4 proteins. The neighbor-joining method was used to analyse the phylogenetic tree using the following LEA protein sequences downloaded from NCBI: MtLEA4-2 XP_003593602.1 (*M. truncatula*) MtLEA4-3 XP_003593603.1 (*M. truncatula*) MtLEA4-1 XP_003590405.1 (*M. truncatula*) MtLEA4-4 4XP_003595694.1 (*M. truncatula*) GmLEA4 AAU94912.1 (*G. max*) GsLEA4 AAU94911.1 (*G. soja*) AhLEA4 APO15550.1 (*A. hypogaea*) AtLEA4-5 NP196294.1 (*A. thaliana*) EgLEA4 AFG26321.1 (*Elaeis guineensis*) GaLEA4 EPS73164.1 (*Genlisea aurea*) ZmLEA4 NP 001150746.2 (*Z. may*) GhLEA4 AAA33057.1 (*Gossypium hirsutum*) JcLEA4 NP 001292972.1(*J. curcas*) SlLEA4 ABA39836.1 (*Solanum lycopersicum*) CaLEA4 All99861.1(*Cicer arietinum*). The bootstrap values (500 replicates) are shown on the branches. (C) Predicted α-helix for amino acid sequence of MsLEA4-4. (D) Predicted hydrophilicity/hydrophobicity for amino acid sequence of MsLEA4-4.

The recombinant plasmid pA7-*MsLEA*4-4-GFP and the empty vector pA7-GFP were transformed to onion epidermal cells through particle bombardment. Using a laser-scanning confocal microscope, we obtained images under dark conditions after 20 h. The empty vector pA7-GFP exhibited a GFP signal throughout the cell wall and the cell nucleus. However, it was detected only in the nuclei of cells transformed with the vector harboring the MsLEA4-4 protein (Fig 2B).

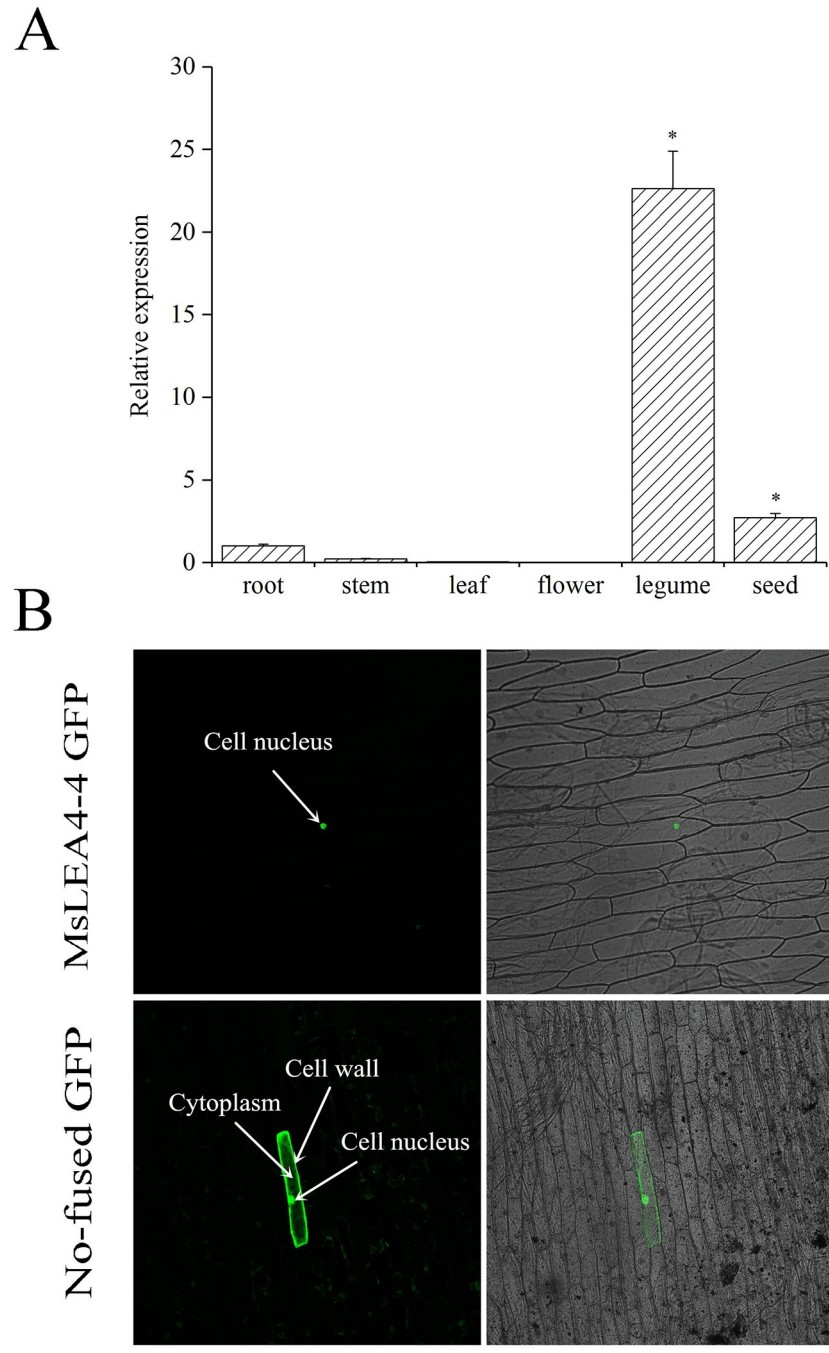

**Fig 2. Expression levels and position of *MsLEA*4-4 in *M. Sativa*.** (A) Expression levels of *MsLEA*4-4 in different tissues of *M. Sativa*. RNA was extracted from different organs of alfalfa. The data were reduplicated three times and showed as mean±SD. (B) Subcellular localization of the MsLEA4-4 protein in onion epidermal cells. (a) Photograph of pA7-*MsLEA*4-4-GFP was taken in a dark field for green fluorescence. (b) Photograph of pA7-*MsLEA*4-4-GFP was taken in a bright light. (c) Photograph of pA7-GFP was taken in a dark field for green fluorescence. (d) Photograph of pA7-GFP was taken in a bright light. The fusion constructor pA7-*MsLEA*4-4-GFP and control plasmid pA7-GFP were transferred into onion epidermal cells by particle bombardment. After 16 h (25˚C), the two transformed cells were observed under a confocal microscope.

## *cis*-element analysis of *MsLEA*4-4 promoter

A 490 bp promoter of *MsLEA*4-4 was cloned with a Genome Walking Kit (Takara) to elucidate the molecular components regulating *MsLEA*4-4 gene expression. The primers used are shown in table (S1 Table). The promoter sequences were analyzed using the Plant CARE program. From that, we found 9 TATA sequences, 3 CAAT enhancer boxes, and 18 putative regulatory *cis*-elements in the promoter. 11, 3, 2, 1, and 1 *cis*-elements were related to light responsiveness, ABA responsiveness, endosperm expression, GA responsiveness, and circadian control (Table 1).

## *MsLEA*4-4 expression levels in alfalfa under phytohormone treatment and light conditions

*MsLEA*4-4 expression levels were examined in alfalfa under different illumination times, photoperiods, and stress durations upon treatment with 0.1 mM ABA and 100 mg/ml GA because of the presence of *cis*-elements in the *MsLEA*4-4 promoter. As shown in Fig 3A and 3B, under ABA and GA treatment, *MsLEA*4-4 expression in alfalfa suffered a steep decline in 2–4 h. The *MsLEA*4-4 expression returned the control levels at 6 h, nudged toward the highest level at 12 h, and moved down to the control level at 48 and 72 h under ABA treatment (Fig 3A). Under GA treatment, the *MsLEA*4-4 expression levels returned to control levels at 24 h, edged up the peak at 48 h, and plunged close to zero at 72 h (Fig 3B). Photoperiod affects plant flowering, growth development and dormancy. Our data showed up that *MsLEA*4-4 expression levels saw a marked increase in response to photoperiods, and the effect of short-day photoperiods overshadowed the long-day (Fig 3C). Constant light and darkness increased *MsLEA*4-4 expression levels; these levels gradually increased with prolonged induction time and reached a maximum at 72 h (Fig 3D). Seedlings were also conditioned to darkness for 72 h and then allowed to recover in light. *MsLEA*4-4 expression at 15 min, 30 min, 2, 12, 24, 48, and 72 h was measured. The data clearly showed that *MsLEA*4-4 expression levels slumped at 15 min and continued to decrease as the duration of light treatment was extended. Specifically, *MsLEA*4-4 expression plummeted at 2 h and then returned to the control level at 12 h. A sharp decline at 72 h came after an upsurge at 24 and 48 h in *MsLEA*4-4 expression (Fig 3E). These data suggested that both the presence and absence of light regulated *MsLEA*4-4 expression.

## Transformation of *Arabidopsis* and phenotypic modulation

pCAMBIA3301 belongs to CaMV 35S promoter, which was utilized to express *MsLEA*4-4 in this study. This 35S:*MsLEA*4-4 recombinant plasmid and the pCAMBIA3301 empty vector

**Table 1. *Cis*-elements of the *MsLEA*4-4 promoter in alfalfa.**

| Site name | Sequence | Function | No. |
|---|---|---|---|
| ABRE | CACGTG | involved in the abscisic acid responsiveness | 3 |
| ATCT-motif | AATCTGATCG | involved in light responsiveness | 1 |
| Box 1 | TTTCAAA | light responsive element | 1 |
| G-Box | CACGTG | involved in light responsiveness | 6 |
| GAG-motif | AGAGAGT | part of a light responsive element | 1 |
| GCN4_motif | TGAGTCA | involved in endosperm expression | 1 |
| GT1-motif | GGTTAA | light responsive element | 1 |
| I-box | CTCTTATGCT | part of a light responsive element | 1 |
| P-box | GCCTTTTGAGT | gibberellin-responsive element | 1 |
| Skn-1_motif | GTCAT | required for endosperm expression | 1 |
| circadian | CAANNNNATC | involved in circadian control | 1 |

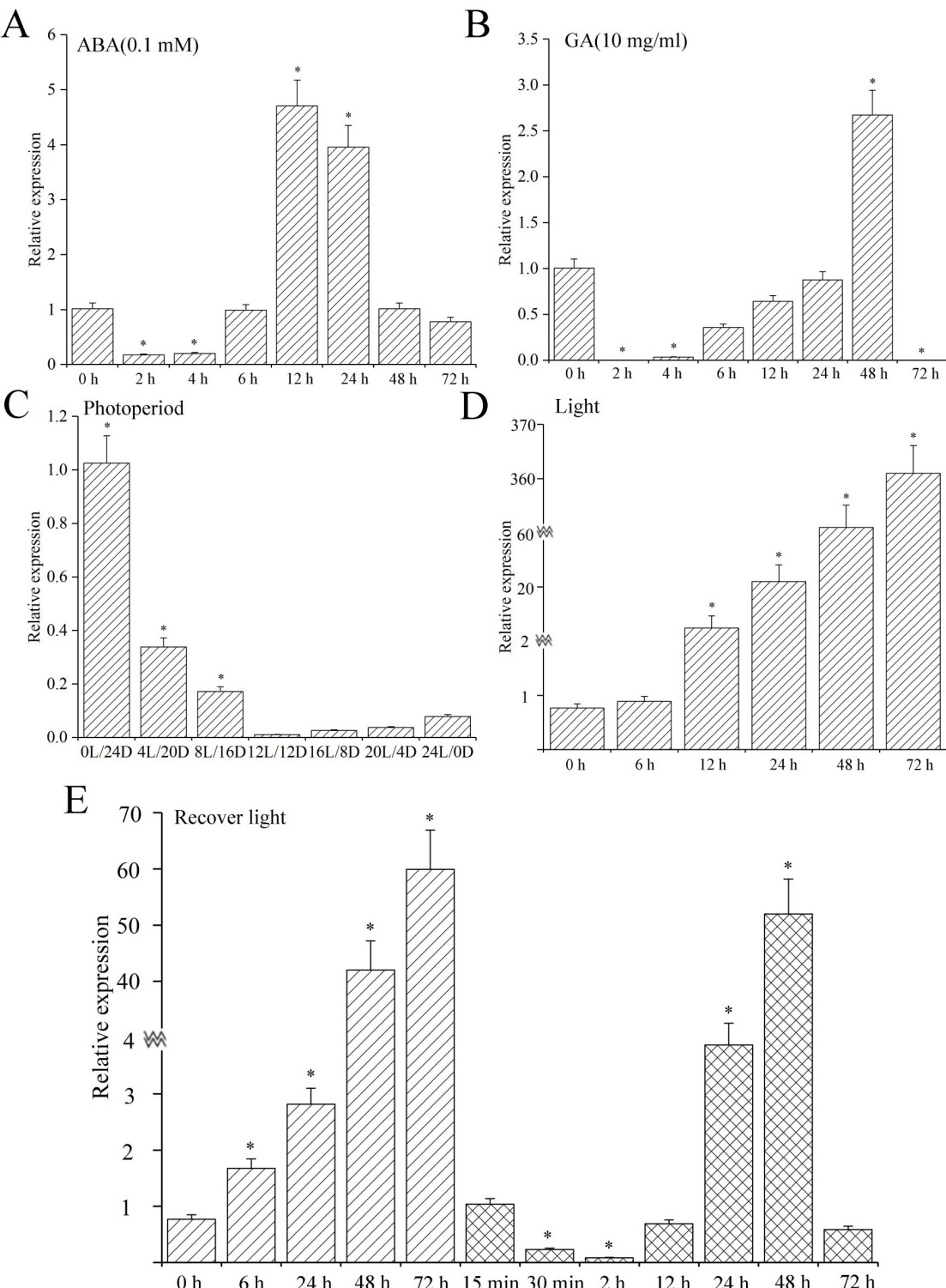

**Fig 3. *MsLEA*4-4 expression in alfalfa under phytohormone treatment and light conditions.** (A) The expression of *MsLEA*4-4 in alfalfa under ABA treatment. (B) The expression of *MsLEA*4-4 in alfalfa under GA treatment. RNA was extracted from 4-week-old alfalfa seedlings treated with 0.1 mM ABA and 10 mg/ml GA at different times. (C) The expression of *MsLEA*4-4 under different photoperiods. Alfalfa seedlings at 4 weeks of age were maintained under different photoperiods, and RNA was extracted after 1 week. (D) The expression of *MsLEA*4-4 under different illumination times. (E) The expression levels of *MsLEA*4-4 in response to dark and light. Four-week-old alfalfa was placed darkness for 72 h then transferred to light conditions to grow, and the RNA of alfalfa in different time was extracted. The data were replicated three times and showed as mean±SD.

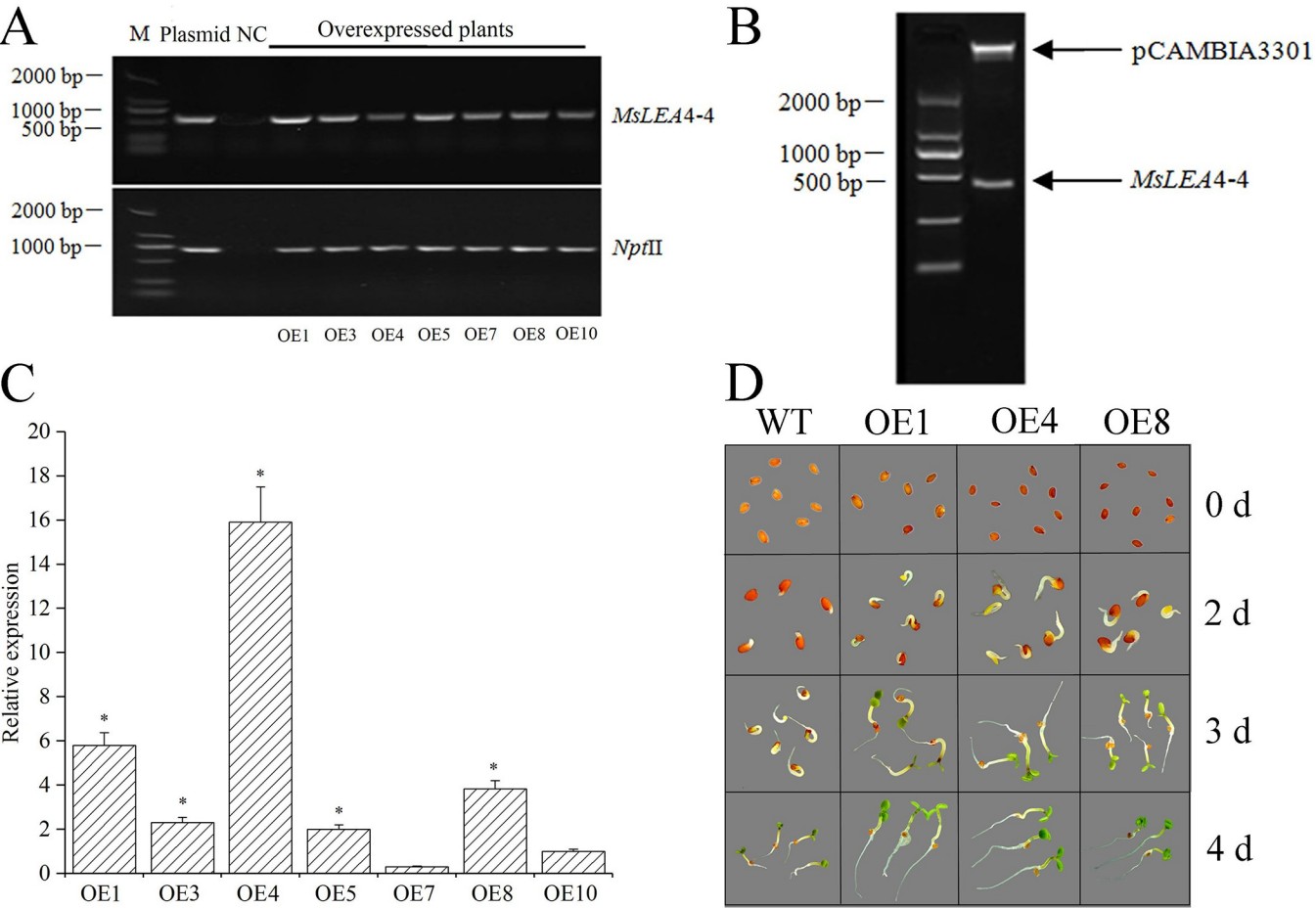

**Fig 4. Molecular and phenotypic detection of *MsLEA4-4* overexpressed plants.** (A) Molecular confirmation of *MsLEA*4-4 overexpressed plants. DNA was extracted from 4-week-old *A. thaliana* seedlings. The PCR primers were from *Npt*II and *MsLEA*4-4 (S1 Table). (B) Enzymatic digestion of the overexpression vector pCAMBIA3301-*MsLEA*4-4. *Bgl*II and *Bste*II were used for the digestion. (C) Expression analysis of *MsLEA*4-4 in 7 overexpressed plants. Total RNA was extracted from both overexpressed plants and WT seedlings aged 4 weeks. The data were replicated three times and showed as mean ± SD. (D) Germination of *MsLEA*4-4-overexpressing plants and WT. The germination of the overexpressed and WT seeds was observed on the 2nd day, the 3rd day and the 4th day.

were transformed into *A. thaliana*. The PCR and qRT-PCR were performed to identify overexpressed plants with the primers for the *Npt*II and *MsLEA*4-4 gene sequences (S1 Table). 7 overexpressed plants were proved to contain *MsLEA*4-4 (Fig 4A). The extracted DNA from 4-week-old overexpressed plants served for *Bgl*II/*Bste*II enzymatic digestion of the overexpression vector pCAMBIA3301-*MsLEA*4-4. Two bands were detected in the electrophoretogram. The uppermost band was pCAMBIA3301, with the lower band of *MsLEA*4-4 (Fig 4B). In the overexpressed plants, *MsLEA*4-4 was inserted into the *A. thaliana*. genome randomly because of different restriction endonucleases sites. In addition, the expression level of *MsLEA*4-4 was either high or low. qRT-PCR was conducted to examine the transcript levels of the introduced *MsLEA*4-4 in *A. thaliana*, and the results confirmed that *MsLEA*4-4 was highly expressed in three overexpressed plants (OE1, OE4, and OE8) (Fig 4C). The *MsLEA*4-4 was confirmed integrated into the genomic DNA of overexpressed plants (OE1, OE4 and OE8) by southern blot analysis. The result showed that there was no hybridization band in WT, while the overexpressed plants (OE1, OE4 and OE8) showed obvious hybridization signals and they were revealed single copy in the genomic DNA (S2 Fig). So the overexpressed plants (OE1, OE4 and

OE8) could use in the subsequent functional experiment. Seed germination was observed for the overexpressed plants harboring *MsLEA*4-4 (OE 1, OE 4, and OE 8) and WT under standard culture conditions was observed. The result was the seeds of overexpressed plants germinated approximately 2 days earlier than the WT (Fig 4D).

## Germination of *MsLEA*4-4 overexpressed plants in response to abiotic stress treatment

The germination rates of the seeds of *MsLEA*4-4-overexpressing *A. thaliana* plants (OE1, OE4, and OE8) and WT were measured under different abiotic stresses and ABA treatment. Each culture dish consisted of 100 seed, and 3 biological repeats. The data showed up that under NaCl stress, the germination rates of overexpressed plants (OE1, OE4, and OE8) reached 37.5%, 41.2%, and 55.8% on the 10th day, respectively, whereas that of the WT was only 11% (Fig 5A). Under ABA treatment, its germination was delayed with a lower germination rate, beginning on the 5th day. As of the 8th day, the germination rate of overexpressed plants reached 59.5%–73.6%, whereas that of the WT was only 6.1% (Fig 5B). The germination of overexpressed plants came 3 days earlier measured against WT under mannitol stress, with the germination rate of 100% on the 5th day. The germination rate of the WT seeds was observed on the 7th day (Fig 5C). To subject seeds to low-temperature stress, a Petri dish was exposed to 4˚C for 10 days and then moved into an illumination incubator (22˚C/18˚C, 16 h light/8 h dark). Notably, WT seeds failed to germinate, whereas the seeds of overexpressed plants started to sprout on the 7th day, and the germination rate reached approximately 50% on the 10th day. After the seeds of overexpressed plants were moved to the illumination incubator, their germination rate reached 100% after 24 h; the WT seeds joined the rapid germination, reaching 100% germination on the 3rd day (Fig 5D). The 4-day-old seedlings of overexpressed plants and WT were photographed under ABA treatment and mannitol stress. The seedlings exposed to 4˚C stress were photographed on the 2nd day after 10 days of chilling, and the seedlings were then transferred to the illumination incubator. The overexpressed seedlings had green cotyledons, whereas the WT seedlings remained at the embryonic growth stage under ABA treatment, mannitol, and 4˚C stress. Under NaCl stress, the overexpressed plants and WT seedlings all grew slowly. By the 10th day, the overexpressed plants seedlings had produced two cotyledons, whereas the WT plants were still at the stage of radicle growth (Fig 5E).

## Lateral root growth of *MsLEA*4-4-overexpressing plants in response to abiotic stress

In this experiment, the lateral root growth of overexpressed plants and WT under different stresses were observed and measured. Seeding was performed on culture dishes containing 1/2 MS. After 72 h of vernalization at 4˚C, the seedlings were transferred to an illumination incubator to grow. As soon as the roots began to grow, the seedings were transferred to culture dishes containing 50 mM NaCl and 150 mM mannitol. Each of them consisted of four species, the WT, OE1, OE4, and OE8, of which each included five seedlings. Lateral root growth was observed and photographed after 5 days. Notably, overexpressed plants under abiotic stress outperformed WT in the lateral root growth capacity, with the most marked difference in plants under both low temperature and NaCl stress (Fig 6A). For each treatment, 25 seedlings were taken, and the lateral roots were counted. This procedure was repeated three times. Under NaCl, 4˚C, mannitol, and 4˚C + NaCl stress, the WT seedlings had 2, 2, 3, and 4 lateral roots, whereas the overexpressed plants had 8, 6, 8, and 8 lateral roots, respectively. These values were remarkably higher than those for WT (Fig 6B). The results indicated that the *MsLEA*4-4 gene could improve the stress of *A. thaliana* through growth of lateral roots.

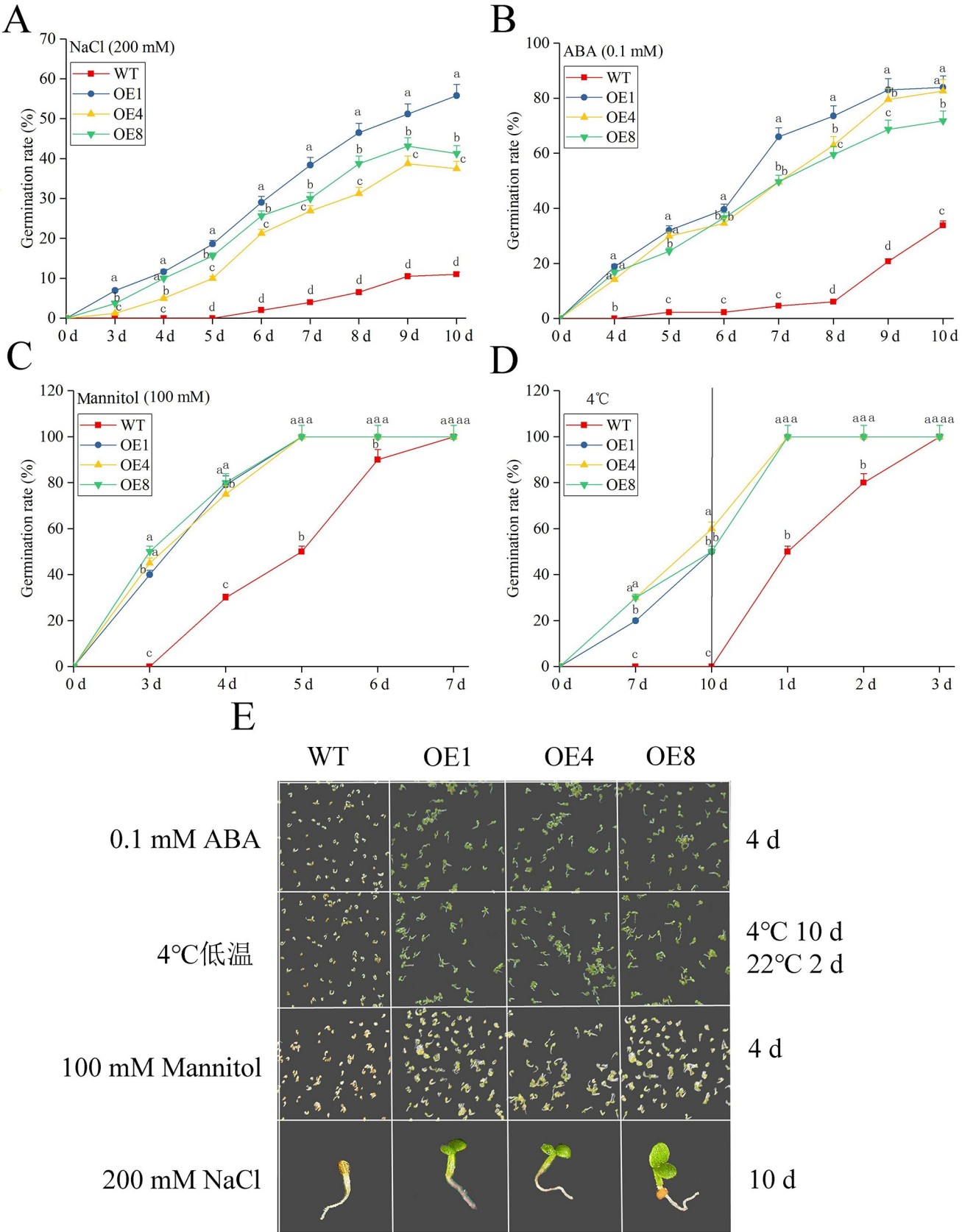

**Fig 5. The germination rates of overexpressed plants (OE1, OE4, and OE8) and WT under abiotic stress and ABA treatment.** (A-D) The germination rates of overexpressed plants (OE1, OE4, and OE8) and WT over 10 days of NaCl (200 mM), 4°C, mannitol (100 mM) stress and ABA (0.1 mM) treatment. All seeds were moved to an illumination incubator (22°C/18°C, 16 h light/8 h dark) to germinate after 72 h of vernalization. For the 4°C condition, the culture dish was maintained at 4°C for 10 days and then moved to the illumination incubator for seed germination. All treatments included three biological replicates. The results are presented as the mean ± SD. The Lowercase letters represent significant differences (*P*<0.05). (E) The growth situation of overexpressed plants and WT under 0.1 mM ABA, 4°C, 100 mM mannitol and 200 mM NaCl on the 4th or 10th day. Photographs of the plants under 4°C, mannitol, NaCl stress and ABA treatment were taken on the 2nd day, the 4th day and the 10th day.

## Leaf color changes in *MsLEA*4-4-overexpressing plants in response to several abiotic stresses

The leaves of the overexpressed plants (OE1, OE4, and OE8) and the WT under different abiotic stresses and ABA treatment were observed and measured. The plants were seeded on culture dishes containing 1/2 MS to grow 7 days, then the seedlings were transferred to 1/2 MS culture medium containing ABA (0.1 mM), $H_2O_2$ (10 mM) and NaCl (300 mM) to grow for 7 days. The results showed that the leaves of the WT were chlorinated or bleached beyond measure in comparison to the overexpressed plants (Fig 7A). The chlorophyll content of overexpressed plants and WT under $H_2O_2$ (10 mM), NaCl (300 mM) and ABA (0.1 mM) treatment, was measured to further clarify the effects of these stresses. Notably, the chlorophyll content decreased in plants under the different stresses. In particular, the chlorophyll content of the overexpressed plants was significantly higher than that of the WT (Fig 7B). NaCl stress contributed to the most obvious bleaching in the WT, and the survival rates of the overexpressed plants and the WT calculated after 7 days of stress were 60%–80% and 20%, respectively (Fig 7C).

## The resistance evaluation of overexpressed plants under osmotic stress

LEA proteins play important roles in plants exposed to abiotic stresses. The experimental data above indicated that the overexpressed plants were superior in germination rates, lateral root numbers and chlorophyll content compared to WT. lateral root numbers and chlorophyll content of the overexpressed plants were superior to those of the WT. Abiotic stresses may induce the expression changes in a series of genes related to osmotic stress in overexpressed plants. In order to demonstrate whether *MsLEA*4-4 transferred into *A. thaliana* could change the expression level of genes related to osmotic stress, some genes such as *NCED*9, *NCED*5, *ABF*3, *ABI*5, *RD29A*, *RAB*18, and *P5CS* were selected. Among these genes, *ABF*3, *ABI*5, *NCED*5, and *NCED*9 exhibited markedly increased transcript levels in three transgenic lines (Fig 8A). To evaluate the stress tolerance of the overexpressed plants, the biomasses of the transgenic and WT plants exposed to abiotic stresses was measured. The overexpressed plants and WT displayed similar biomasses under normal conditions, but the overexpressed plants exhibited a higher biomass compared with WT under abiotic stress conditions (Fig 8B).

## *MsLEA*4-4 overexpressing improved antioxidant enzyme activity in transgenic *Arabidopsis* under abiotic stress and ABA treatment

Under abiotic stress, plants can adjust intracellular osmotic pressure and stabilize the intracellular activity of enzymes to enhance resistance through synthesis and accumulation of soluble sugars and Pro. In our study, the soluble sugar, Pro, and MDA content, the CAT, POD, and SOD activity of the overexpressed plants and the WT were measured under different abiotic stresses and ABA treatment. Under different stresses, both overexpressed plants and WT presented an increase in the soluble sugar content. In particular, the soluble sugar content of the overexpressed plants was higher than that of the WT (*P*<0.05) under all abiotic stresses except

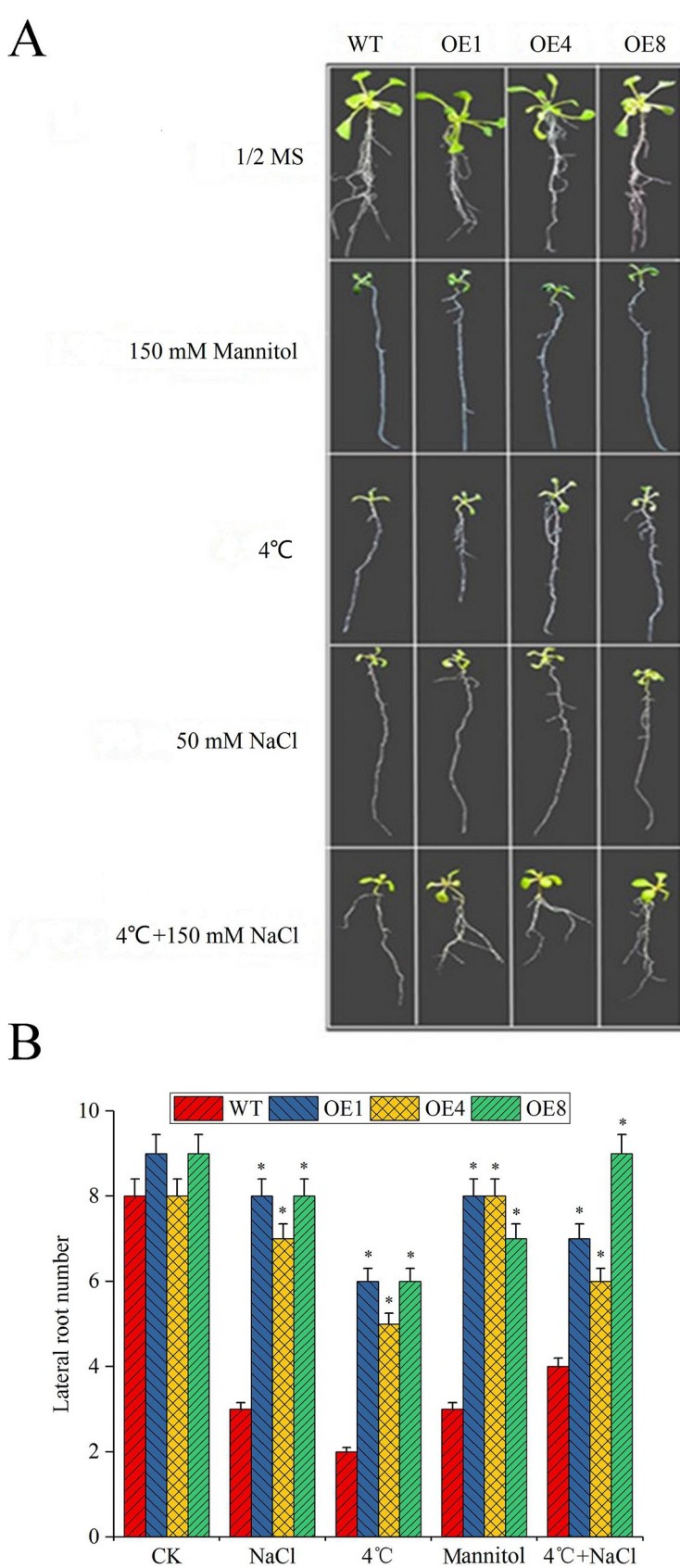

**Fig 6. Lateral root growth of *MsLEA*4-4-overexpressing *Arabidopsis* (OE1, OE4, and OE8) and WT in response to abiotic stress treatment.** The abiotic stress treatments were mannitol (150 mM), 4˚C, NaCl (50 mM), and 4˚C + NaCl (150 mM). (A) Comparison of the lateral roots of overexpressed plants and WT after 5 days of abiotic stress. (B) Lateral root numbers of overexpressed plants and WT. The NaCl concentration was 50 mM in the experiment, because it was observed that there was slight difference in the numbers of lateral roots of overexpressed plants and WT under concentrations of 100, 150, 200, 250, and 300 mM.

NO. The soluble sugar content in the overexpressed plants was more obvious than that in the WT under $H_2O_2$ and NaCl stress (Fig 9A). The CAT activity of the overexpressed plants was also significantly higher ($P<0.05$) than that of the WT under abiotic stress. The mannitol, $H_2O_2$, ABA, NaCl stress and ABA treatment data revealed obvious changes. Under $H_2O_2$ stress, the CAT activity of the overexpressed plants was 92.4% higher than that of the WT (Fig 9B). When plants are exposed to abiotic stress, PODs remove oxygen from cells and produce

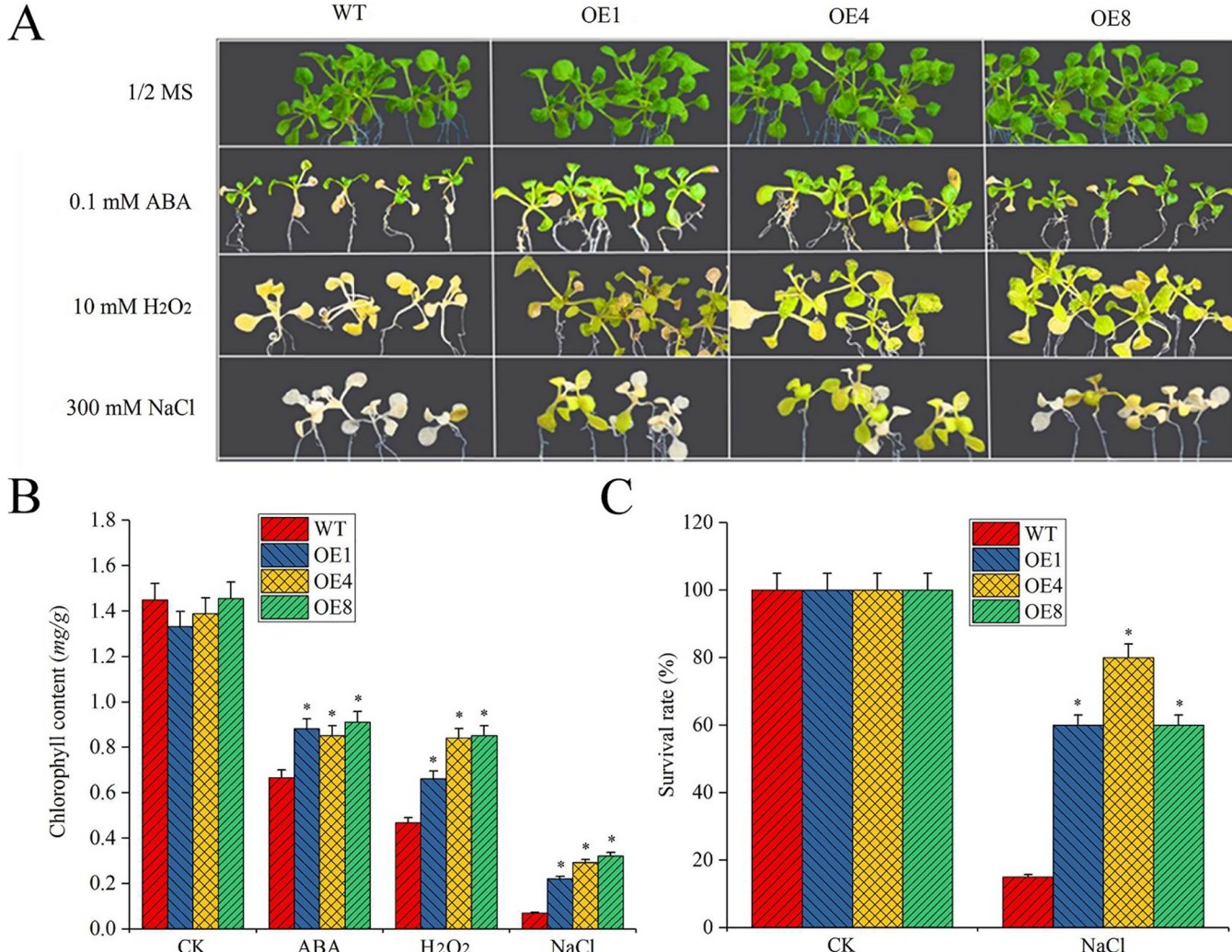

**Fig 7. The comparison of overexpressed plants (OE1, OE4, and OE8) and WT in response to abiotic stress.** (A) Leaf color of overexpressed plants (OE1, OE4, and OE8) and WT in response to NaCl, $H_2O_2$ stress and ABA treatment for 7 days. (B) The chlorophyll content of overexpressed plants and WT under NaCl, $H_2O_2$ stress and ABA treatment for 7 days. The abiotic stresses were $H_2O_2$ (10 mM), NaCl (300 mM) and ABA (0.1 mM) treatment. (C) The survival rates of overexpressed plants and WT under 300 mM NaCl treatment for 7 days. Twenty seedlings acted as one replicate, and each treatment was performed with three biological replicates. The data in the figure were repeated three times and showed as mean ± SD.

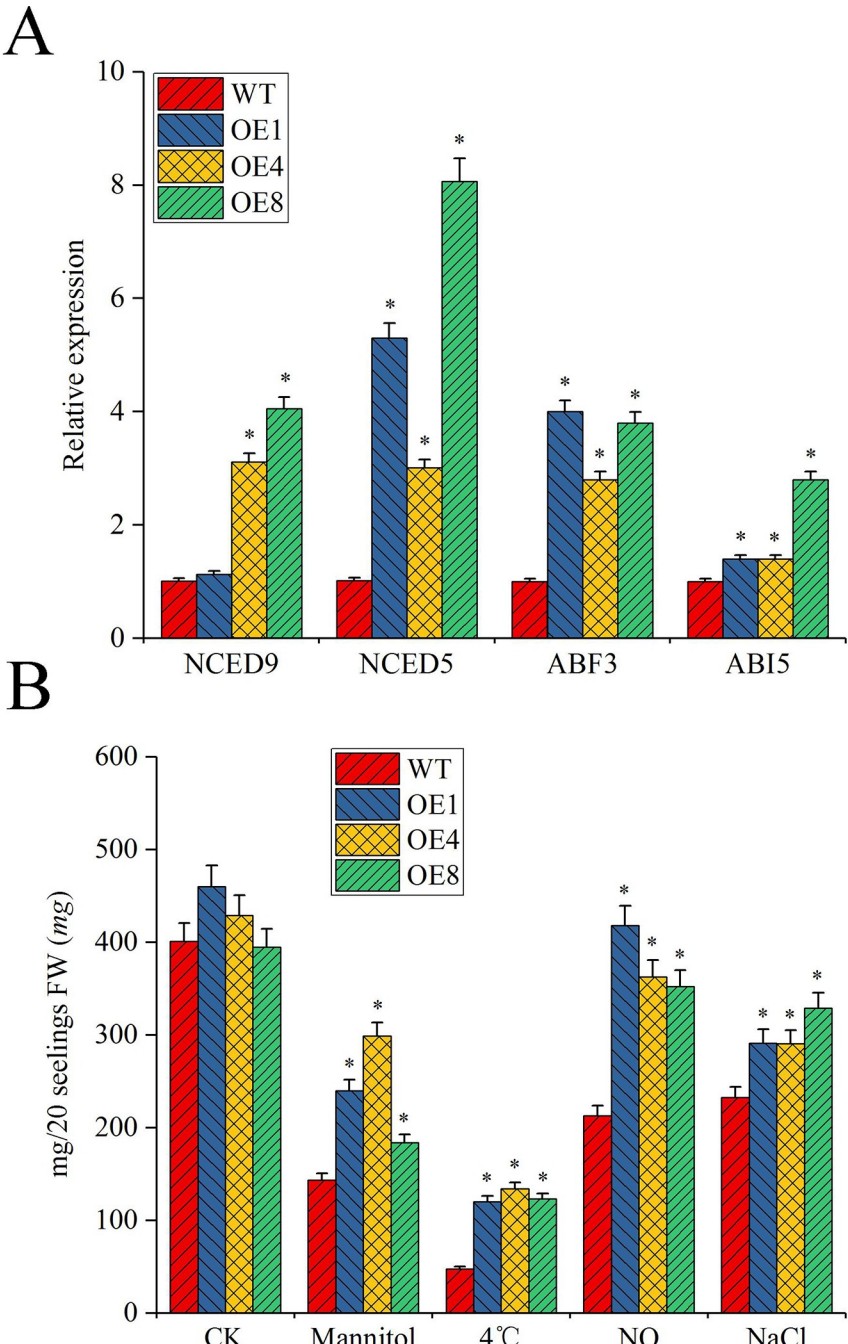

**Fig 8. Expression levels of several osmotic related genes in the overexpressed plants.** (A) Expression levels of several osmotic stress related genes in the overexpressed plants (OE1, OE4, and OE8) and WT under mannitol, 4°C, NO and NaCl treatment. Both the overexpressed plants (OE1, OE4, and OE8) aged one week and WT were treated with 100 mM mannitol, 4°C, 0.1 mM NO or 200 mM NaCl for 7 days. RNA was extracted from plants under all treatments, pooled for each treatment and used for expression analysis. (B) Biomass of the overexpressed plants (OE1, OE4, and OE8) and the WT under mannitol, 4°C, NO and NaCl treatment. One-week-old *Arabidopsis* seedlings were treated with 100 mM mannitol, 4°C, 0.1 mM NO or 200 mM NaCl for 7 days and harvested. Each treatment was replicated three times, and there were twenty seedlings in each replicate. The data were showed as mean ± SD.

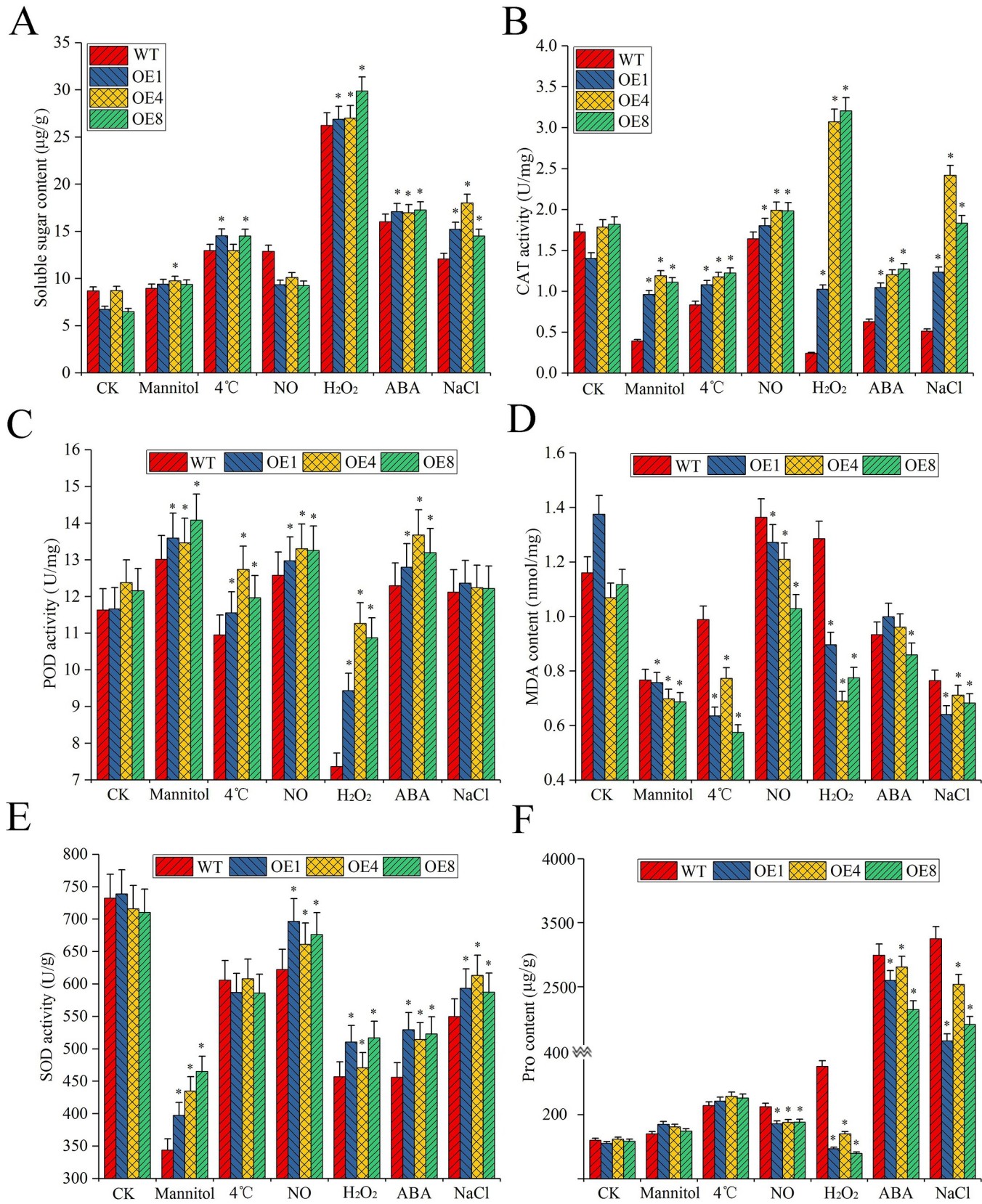

**Fig 9. Antioxidant enzyme activity in the overexpressed plants and the WT under abiotic stress and ABA treatment.** (A-F) The soluble sugar content; CAT, POD, MDA, and SOD activity; and the Pro content of overexpressed plants (OE1, OE4, and OE8) and WT under abiotic stress. Both 1-week-old overexpressed plants and WT were treated with 200 mM mannitol, 4°C, 0.1 mM NO, 10 mM $H_2O_2$, 200 mM NaCl or 0.1 mM ABA for 7 days to detect antioxidant enzyme activity. Each treatment was composed of 0.1 g of seedlings and repeated three times.

some of the metabolites necessary for cells. The POD activity was higher in the overexpressed plants than that in the WT under different stresses, with a difference of 34.6% under $H_2O_2$ stress ($P<0.05$). (Fig 9C). MDA content is positively correlated with changes in membrane permeability. Plant tissues or organelle membranes undergo peroxidation reactions when the plants are exposed to abiotic stress, thereby producing high levels of MDA. In this study, MDA content in the overexpressed plants was much lower ($P<0.05$) compared to WT under the abiotic stresses except NO stress and ABA treatment, with a marked difference of 53.7% under $H_2O_2$ stress (Fig 9D). SOD can eliminate harmful substances produced during the metabolism. The data revealed that the there was a decrease of the SOD activity both in the overexpressed plants and the WT under abiotic stress, while the overexpressed plants exhibited relatively ($P<0.05$) higher activity, except under cold stress (Fig 9E). Pro, as one of the most effective osmotic substances in plants, reflects the resistance of plants to abiotic stress. The Pro content of the overexpressed plants and the WT shot up under NaCl stress and ABA treatment, but relatively the former was 68.4% and 34.5% lower respectively under these stresses, and 3.43-fold lower under $H_2O_2$ stress (Fig 9F). These results indicated that the overexpressed plants showed more tolerant to salt, drought, cold, and oxidative stress than the WT, and also can be induced by ABA treatment.

## Discussion

LEA proteins, as drought stress–induced proteins among a large family in plants, are involved in plant development, and studies have found that different LEA proteins are highly hydrophilic and heat stable, lack cysteine and tryptophan residues and are rich in glycine or other amino acids, such as alanine, serine, and threonine [47]. Proline and glycine belong to helical breaking proteins, while alanine, glutamic acid and lysine are in non-helical breaking proteins. The content of helical forming protein is very high, while helical breaking protein is little in LEA4, so most LEA4 shows high α-helical [48]. In this study, MsLEA4-4 protein was found to be deficient in cysteine and tryptophan and rich in glycine, alanine, glutamine, lysine, arginine, and threonine; in addition, the protein contains lots of α-helix (Fig 1C). The α-helix usually has a signal transduction or transport function in plant cells [49], which may enhance the ability to abiotic stress of MsLEA4-4-overexpression plants.

LEA4 was located in many position of the cell, and several LEA4 hade TAQAAKEKAXE sequence, while others do not [33]. Proteins located in the nucleus may have transcriptional and ABA responsive functions. The LEA4 from *Brassica napus* was located in the nucleus and have 25% α-helix [50]. In this study, subcellular localization showed *MsLEA*4-4 localization in the nucleus (Fig 2B) and had five repeated 11-mer sequences of TAQAAKEKTQQ (Fig 1A). The structure of *MsLEA*4-4 well contributes to its stress resistance.

When plants encountered abiotic stresses, such as drought, high salt, and low temperature, cellular osmotic regulation changed because of water deficiency, which would induce the regulation and expression of a series of genes related to abiotic stress, and promoters feature prominently in these processes [51]. ABA, a signaling molecule, participates in plant adaptation to abiotic stresses such as drought, high salt, and low temperature and in seed maturation and dormancy [52]. The promoter regions of most genes contain ABA-acting elements [53]. *LEA* promoter regions generally consist of ABA-responsive elements (ABREs), which can initiate

the transcription of genes responsive to ABA by binding to the core motif (ACGT) of transcription factors. In addition, the promoter regions of *LEA* genes contain *cis*-acting elements related to ABA pathways, such as G-box (CACGTC) and A-box (TACGTA) elements [54]. *LEA*4 can be induced under ABA treatment, such as *AtLEA*4-5 and *BnLEA*4 [55]. In this study, the promoter region of *MsLEA*4-4 was found to contain three ABREs and two endosperm expression motifs (TGAGTCA and GTCAT). Studies have shown that *MsLEA*4-4 expression presents a parabolic trend during ABA treatment (Fig 3A), indicating that *MsLEA*4-4 is sensitive to ABA and plays a potential role in regulating the stress resistance of alfalfa through the ABA signaling pathway in vivo.

Moreover, the *MsLEA*4-4 promoter region comprised 11 *cis*-acting elements related to light induction and 1 *cis*-acting element related to the biological clock (Table 1). The LEA content in alfalfa increased under extended dark stress and light stress (Fig 3D and 3E). Photoperiod gives an influence on plant flowering, growth development, and dormancy. The data clearly showed that *MsLEA*4-4 expression levels increased significantly in response to short-day and long-day photoperiods, with stronger signs of the former (Fig 3C). The leaf structure will be destroyed and photosynthetic rate will be declined under abiotic stress. Eventually, the biomass of the plants decreased [56]. A variety of *cis*-acting elements related to light in LEA may protect leaf structure and photosynthetic system, and guarantee the biomass of plant. The results showed that the chlorophyll content and biomass were significantly higher in overexpressed plants than in WT under abitic stress (Fig 7B).

LEA4 proteins play important roles in plant responses to external stress. Previous studies have reported the functions of LEA4 proteins related to plant stress resistance [57]. Overexpression of *LEA* 4 in *Arabidopsis thaliana*, the root and stem length, and the survival rate of overexpressed plants was improved compared to wild type plants [50]. *MsLEA*4-4 from *M. sativa* has been successfully transferred into the *A. thaliana* ecotype Columbia-0. Our results showed that germination of *MsLEA*4-4-overexpressing seeds occurred 1–2 days earlier than measured against WT under different stresses (Fig 5E). The germination rates (Fig 5A–5D) and the numbers of lateral roots (Fig 6B) of the overexpressed plants were higher than those of the WT. The overexpressed plants hold a remarkable dominance of salt tolerance, with survival rates reaching 60%–80%, whereas the survival rate of the WT was only 20% under salt stress (Fig 7C). Under abiotic stress, LEA proteins may protect the antioxidant activity in plants to enhance resistance to drought and various stresses. A previous study found that the MDA content in *LEA*-transgenic *Salvia miltiorrhiza* was low under drought and high-salt conditions and their SOD activity and glutathione concentrations were higher compared with the control plants [58]. In our study, mannitol, high salt, low temperature, $H_2O_2$, and other stresses were applied. Our results revealed that the soluble sugar content (Fig 9A) and the CAT (Fig 9B), POD (Fig 9C), and SOD (Fig 9E) enzymatic activity were higher in the overexpressed plants than in the WT. When plants encounter abiotic stress, the oxidative degradation process is inhibited, and a consequent increase in Pro content can add protection for plant growth. However, the higher Pro contents in the plant, the greater the damage [59]. The Pro (Fig 9F) and MDA levels in the overexpressed plants were lower than those in the WT under different stresses, and the enhanced resistance of the overexpressed plants was most obvious under NaCl and $H_2O_2$ stress (Fig 9D). The results showed that *MsLEA*4-4-overexpression *A. thaliana* plants were less damaged than WT under abiotic stress, and reduced the damage of abiotic stress through a variety of antioxidant enzyme system to balance the ROS of overexpressed plants.

These results suggest that transfer of *MsLEA*4-4 into *A. thaliana* enhances the resistance of the overexpressed plants to abiotic stresses, such as drought, high salt, low temperature, and oxidative stress. The gene expression analysis results indicate that *MsLEA*4-4 may improve the resistance of overexpressed plants by regulating the expression of a series of genes related to

osmotic stress in plants. Thus, *MsLEA*4-4 transferred into alfalfa may join the regulation of stress and improve the resistance of alfalfa. Further studies should be performed in which *MsLEA*4-4 is transferred into alfalfa, to investigate the function of *MsLEA*4-4 in alfalfa and to develop *MsLEA*4-4-overexpressing alfalfa varieties with strong resistance to stress.

## Supporting information

**S1 Table. Primers used in the study.**
(DOC)

**S1 Fig. Constructing expression vector of *MsLEA*4-4.** (A) Diagram of the *MsLEA*4-4 overexpression vector. (B) Diagram of the *MsLEA*4-4-GFP expression vector.
(DOC)

**S2 Fig. Southern blot analysis of *MsLEA*4-4 overexpressed plants (OE1, OE4 and OE8).**
(DOC)

**S1 Raw Images.**
(PDF)

**S2 Raw Images.**
(PDF)

## Acknowledgments

We would like to grateful to Dr. Xuemin Wang for helpful comments on an early version of this manuscript. We thank Dr. Yonghong Shi for funding to this study.

## Author Contributions

**Conceptualization:** Huili Jia, Xuemin Wang, Kuanhu Dong.

**Data curation:** Huili Jia, Xuemin Wang, Zhihong Fang, Kuanhu Dong.

**Funding acquisition:** Yonghong Shi, Kuanhu Dong.

**Investigation:** Huili Jia, Xuemin Wang, Zhihong Fang.

**Methodology:** Huili Jia, Xuemin Wang, Chunyan Li.

**Project administration:** Kuanhu Dong.

**Resources:** Yunqi Wang, Jianning Liu, Kuanhu Dong.

**Supervision:** Yunqi Wang.

**Validation:** Kuanhu Dong.

**Visualization:** Huili Jia, Xuemin Wang, Xinming Wu.

**Writing – original draft:** Huili Jia, Xuemin Wang, Zhihong Fang, Chunyan Li, Kuanhu Dong.

**Writing – review & editing:** Huili Jia, Xuemin Wang, Kuanhu Dong.

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
