## [Decision Letter · Decision Letter 0]

18 Feb 2020

PONE-D-19-31989

Overexpression of Medicago sativa LEA4-4 can improve the salt, drought, and oxidation resistance of transgenic Arabidopsis

PLOS ONE

Dear Mrs. Huili,

Thank you for submitting your manuscript to PLOS ONE. After careful consideration, we feel that it has merit but does not fully meet PLOS ONE’s publication criteria as it currently stands. Therefore, we invite you to submit a revised version of the manuscript that addresses the points raised during the review process.

We would appreciate receiving your revised manuscript by Apr 03 2020 11:59PM. To enhance the reproducibility of your results, we recommend that if applicable you deposit your laboratory protocols in protocols.io, where a protocol can be assigned its own identifier (DOI) such that it can be cited independently in the future. For instructions see: http://journals.plos.org/plosone/s/submission-guidelines#loc-laboratory-protocols

We look forward to receiving your revised manuscript.

Kind regards,

Keqiang Wu, Ph.D

Academic Editor

PLOS ONE

Journal Requirements:

"This work was supported by a grant from the Special Funds for the Technical System of the Modern Agricultural Industry of China (CARS-34), by the Species Resources Conservation of China (2130135) and by the Key Research and Development Projects of ShanXi (201703D211002-9-2)."

 "Author Contributions

Conceptualization: Huili Jia, Xuemin Wang, Kuanhu Dong

Data curation: Huili Jia, Xuemin Wang, Zhihong Fang, Kuanhu Dong

Funding acquisition: Kuanhu Dong, Yonghong Shi

Investigation: Huili Jia, Xuemin Wang, Zhihong Fang

Methodology: Huili Jia, Xuemin Wang, Chunyan Li

Project administration: Kuanhu Dong

Resources: Jianning Liu, Yunqi Wang, Kuanhu Dong

Supervision: Kuanhu Dong, Yunqi Wang

Validation: Kuanhu Dong

Visualization: Huili Jia, Xuemin Wang, Xinming Wu

Writing-original draft: Huili Jia, Xuemin Wang, Yonghong Shi, Zhihong Fang, Chunyan Li, Kuanhu Dong

Writing-review & editing: Huili Jia, Xuemin Wang, Kuanhu Dong"

Reviewers' comments:

Reviewer's Responses to Questions

**Comments to the Author**

1. Is the manuscript technically sound, and do the data support the conclusions?

Reviewer #1: Yes

Reviewer #2: Partly

Reviewer #3: Yes

2. Has the statistical analysis been performed appropriately and rigorously? 

Reviewer #1: Yes

Reviewer #2: No

Reviewer #3: Yes

3. Have the authors made all data underlying the findings in their manuscript fully available?

Reviewer #1: Yes

Reviewer #2: No

Reviewer #3: Yes

4. Is the manuscript presented in an intelligible fashion and written in standard English?

Reviewer #1: Yes

Reviewer #2: No

Reviewer #3: Yes

5. Review Comments to the Author

Reviewer #1: The manuscript "Overexpression of Medicago sativa LEA4-4 can improve the salt, drought, and oxidation resistance of transgenic Arabidopsis" by Jia et al on the investigation of the LEA genes provides more information on the role of the LEAs, though i wish to draw to the attention of the authors of a more recently published work by Magwanga et al on the LEA genes, it would be good if they incorporate the already known information and very current literature on the introduction and discussion section.

Minor revision is needed and the language requires to be polished. Moreover, it would be good if the authors replace the term "transgenic plants" with the "overexpressed plants"

line 59 "LEA proteins are key proteins involved in cell antistress", delete the repeated protein, ...LEAs are key proteins involved in cell antistress

line 65-66 "With the development of modern biological technologies, LEA proteins in various species have been identified introduce citation

line 113: I doubt how one can express hours in 0.25, 0.5, adopt uniform format, either in hours or minutes

Reviewer #2: 1. The sentence formation in entire manuscript was very poor and it wont fit to get it published in a reputed journal PLOSONE

2. I was surprise that, authors used simple restriction digestion of genomic DNA extracted from putative transgenics rather performing Southern Hybridization.

3. It is highly impossible to elute integrated transgene alone from transgenic plants in Fig.5.b. Suggest to conduct Southern hybridization

Reviewer #3: The reviewer has finished the review of the manuscript entitled “Overexpression of Medicago sativa LEA4-4 can improve the salt, drought, and oxidation resistance of transgenic Arabidopsis“. The manuscript presents an interesting story about Medicago LEA4-4 gene in abiotic stress resistance in transgenic Arabidopsis. The authors found that overexpression of Medicago LEA4-4 exhibits higher survival rate under salt stress condition. In addition, more lateral roots and higher chlorophyll content were observed in the transgenic plants. Although the authors had made revisions based on reviewers’ comments, there are revisions needed for current version.

1. In the Materials and methods section, “two tails tests” was used for statistical analysis. Is T-test was actually used? Please specify correctly.

2. Phytohormone ABA, in fact, should not be treated as stress. Please rephrase throughout the whole manuscript text.

3. In line 226, title “Analysis and expression of the MsLEA4-4 promoter” is not correct and should be rephrased. Promoter-GUS should be included for promoter analysis. For just gene expression, promoter is not needed.

4. Figure 1, the vector information is suggested to be moved to Supplementary files.

5. In Figure 2A, is there any known domain (i.e. TAQAAKEKTQQ amino acid motifs) about LEA can be marked for LEA4-4?

6. In Figure 2B, why the authors only include one MsLEA to be analyzed? How about other group IV LEAs?

7. In Figure 3B, the authors should include organelle markers for imaging.

8. In Figure 4A-D, please mark the treatment in each figure individually.

9. 5D, please label which samples were analyzed for comparison.

10. In Figure 6E, please label which samples were analyzed for comparison.

11. Statistics is needed for Figure 6A-D.

12. Please cite the following two recently published papers related to ABF3 and ABI5 for stress response in the references.

1. Chang HC, Tsai MC, Wu SS, Chang IF. 2019. Regulation of ABI5 expression by ABF3 during salt stress responses in Arabidopsis thaliana. Bot Stud. 60(1):16.

2. Zhang H, Liu D, Yang B, Liu WZ, Mu B, Song H, Chen B, Li Y, Ren D, Deng H, Jiang YQ. 2020. Arabidopsis CPK6 positively regulates ABA signaling and drought tolerance through phosphorylating ABA-responsive element-binding factors. J Exp Bot. 71(1):188-203.

13. It is questioned why proline level decreased in OE lines in Figure 10F. Please provide explanation in the discussion.

14. Please cite the following paper about group 4 LEAs and discuss it in the Discussion section.

Dalal M, Tayal D, Chinnusamy V, Bansal KC. 2009. Abiotic stress and ABA-inducible Group 4 LEA from Brassica napus plays a key role in salt and drought tolerance. J Biotechnol. 139(2):137-45.

6. PLOS authors have the option to publish the peer review history of their article (what does this mean?). If published, this will include your full peer review and any attached files.

Reviewer #1: Yes: Liu Fang

Reviewer #2: No

Reviewer #3: No

---

## [Author Response · Author response to Decision Letter 0]

30 Apr 2020

Reviewer #1: 

The manuscript "Overexpression of Medicago sativa LEA4-4 can improve the salt, drought, and oxidation resistance of transgenic Arabidopsis" by Jia et al on the investigation of the LEA genes provides more information on the role of the LEAs, though I wish to draw to the attention of the authors of a more recently published work by Magwanga et al on the LEA genes, it would be good if they incorporate the already known information and very current literature on the introduction and discussion section.

√ Thank you for your comments. We have read the article of LEA genes recently published by Magwanga et al and added some information and literature into the introduction and discussion section in revised MS. Please check it. 

Minor revision is needed and the language requires to be polished. Moreover, it would be good if the authors replace the term "transgenic plants" with the "overexpressed plants"

√ Thank you for your comments. we have replaced the "transgenic plants" with "overexpressed plants" in revised MS. Please check it.

line 59 "LEA proteins are key proteins involved in cell antistress", delete the repeated protein, ...LEAs are key proteins involved in cell antistress

√ Thank you for your comments. We have revised these sentences according to your advice in revised MS. Please check it.

line 65-66 "With the development of modern biological technologies, LEA proteins in various species have been identified introduce citation

√ Thank you for your comments. We have added the citation according to your advice in revised MS. Please check it.

line 113: I doubt how one can express hours in 0.25, 0.5, adopt uniform format, either in hours or minutes

√ Thank you for your comments. The uniform format was used in revised MS according to your advice. Please check it.

Reviewer #2: 

1. The sentence formation in entire manuscript was very poor and it wont fit to get it published in a reputed journal PLOSONE

√ Thank you for your comments. We have revised the whole sentences according to PLOS ONE's style requirements in revised MS. Also we have revised the whole manuscript carefully and tried to avoid any grammar or syntax error in editing service (http://essaystar.com/service.html). Please check it. 

2. I was surprise that, authors used simple restriction digestion of genomic DNA extracted from putative transgenics rather performing Southern Hybridization.

√ Thank you for your comments. We carried out the Southern Hybridization experiment according to your advice and the results can be found in revised MS (S2 Fig). Please check it.

3. It is highly impossible to elute integrated transgene alone from transgenic plants in Fig.5.b. Suggest to conduct Southern hybridization

√ Thank you for your comments. We carried out the Southern Hybridization experiment according to your advice and the results can be found in revised MS (S2 Fig). Please check it.

Reviewer #3: 

The reviewer has finished the review of the manuscript entitled “Overexpression of Medicago sativa LEA4-4 can improve the salt, drought, and oxidation resistance of transgenic Arabidopsis“. The manuscript presents an interesting story about Medicago LEA4-4 gene in abiotic stress resistance in transgenic Arabidopsis. The authors found that overexpression of Medicago LEA4-4 exhibits higher survival rate under salt stress condition. In addition, more lateral roots and higher chlorophyll content were observed in the transgenic plants. Although the authors had made revisions based on reviewers’ comments, there are revisions needed for current version.

1. In the Materials and methods section, “two tails tests” was used for statistical analysis. Is T-test was actually used? Please specify correctly.

√ Thank you for your comments. The T-test was actually used in MS and we have corrected this error in revised MS. Please check it.

2. Phytohormone ABA, in fact, should not be treated as stress. Please rephrase throughout the whole manuscript text.

√ Thank you for your comments. We have corrected this error in whole MS. Please check it.

3. In line 226, title “Analysis and expression of the MsLEA4-4 promoter” is not correct and should be rephrased. Promoter-GUS should be included for promoter analysis. For just gene expression, promoter is not needed.

√ Thank you for your comments. The title has been rephrased according to your advice. Please check it.

4. Figure 1, the vector information is suggested to be moved to Supplementary files.

√ Thank you for your comments. We have placed the vector information into Supplementary files (S1 Table). Please check it. 

5. In Figure 2A, is there any known domain (i.e. TAQAAKEKTQQ amino acid motifs) about LEA can be marked for LEA4-4?

√ Thank you for your comments. The five conserved domain have been marked in new Figure (Fig 1A) in revised MS. Please check it. 

6. In Figure 2B, why the authors only include one MsLEA to be analyzed? How about other group IV LEAs?

√ Thank you for your comments. The Fig 2B has been reworked and the new Figure (Fig 1B) has been presented in revised MS. Please check it. 

7. In Figure 3B, the authors should include organelle markers for imaging.

√ Thank you for your comments. The organelle markers has been added in the new Figure (Fig 2B) according to your advice in revised MS. Please check it. 

8. In Figure 4A-D, please mark the treatment in each figure individually.

√ Thank you for your comments. We have marked the treatment in each figure (Fig 3A-D) individually according to your advice in revised MS. Please check it. 

9. Figure 5D, please label which samples were analyzed for comparison.

√ Thank you for your comments. We have marked the samples which were analyzed for comparison according to your advice in the new Figure (Fig 4D) in revised MS. Please check it.

10. In Figure 6E, please label which samples were analyzed for comparison.

√ Thank you for your comments. We have labeled the samples which were analyzed for comparison in the new Figure (Fig 5E) according to your advice in revised MS. Please check it.

11. Statistics is needed for Figure 6A-D.

√ Thank you for your comments. We have reworked the Figure 6A-D and added significant difference in the new Figure (Figure 5A-D) according to your advice in revised MS. Please check it.

12. Please cite the following two recently published papers related to ABF3 and ABI5 for stress response in the references.

1. Chang HC, Tsai MC, Wu SS, Chang IF. 2019. Regulation of ABI5 expression by ABF3 during salt stress responses in Arabidopsis thaliana. Bot Stud. 60(1):16.

2. Zhang H, Liu D, Yang B, Liu WZ, Mu B, Song H, Chen B, Li Y, Ren D, Deng H, Jiang YQ. 2020. Arabidopsis CPK6 positively regulates ABA signaling and drought tolerance through phosphorylating ABA-responsive element-binding factors. J Exp Bot. 71(1):188-203.

√ Thank you for your comments. We have cited these papers according to your advice in revised MS. Please check it.

13. It is questioned why proline level decreased in OE lines in Figure 10F. Please provide explanation in the discussion.

√ Thank you for your comments. We have explained the reason why proline level decreased in OE lines in the new Figure 9F in revised MS. Please check it.

14. Please cite the following paper about group 4 LEAs and discuss it in the Discussion section.

Dalal M, Tayal D, Chinnusamy V, Bansal KC. 2009. Abiotic stress and ABA-inducible Group 4 LEA from Brassica napus plays a key role in salt and drought tolerance. J Biotechnol. 139(2):137-45.

√ Thank you for your comments. We have cited this paper according to your advice in revised MS. Please check it.

Thank you again for your kind help and excellent suggestions for our manuscript. We hope these revisions will be satisfactory and will lead to acceptance for publication. We are looking forward to hearing from you soon.

Dr. Kuan-Hu Dong

2020-4-28

---

## [Decision Letter · Decision Letter 1]

19 May 2020

Overexpression of Medicago sativa LEA4-4 can improve the salt, drought, and oxidation resistance of transgenic Arabidopsis

PONE-D-19-31989R1

Dear Dr. Huili,

We are pleased to inform you that your manuscript has been judged scientifically suitable for publication and will be formally accepted for publication once it complies with all outstanding technical requirements.

With kind regards,

Keqiang Wu, Ph.D

Academic Editor

PLOS ONE

Additional Editor Comments (optional):

Reviewers' comments:

Reviewer's Responses to Questions

**Comments to the Author**

1. If the authors have adequately addressed your comments raised in a previous round of review and you feel that this manuscript is now acceptable for publication, you may indicate that here to bypass the “Comments to the Author” section, enter your conflict of interest statement in the “Confidential to Editor” section, and submit your "Accept" recommendation.

Reviewer #1: All comments have been addressed

Reviewer #2: All comments have been addressed

Reviewer #3: All comments have been addressed

2. Is the manuscript technically sound, and do the data support the conclusions?

Reviewer #1: Yes

Reviewer #2: Yes

Reviewer #3: Yes

3. Has the statistical analysis been performed appropriately and rigorously? 

Reviewer #1: Yes

Reviewer #2: Yes

Reviewer #3: Yes

4. Have the authors made all data underlying the findings in their manuscript fully available?

Reviewer #1: Yes

Reviewer #2: Yes

Reviewer #3: Yes

5. Is the manuscript presented in an intelligible fashion and written in standard English?

Reviewer #1: Yes

Reviewer #2: Yes

Reviewer #3: Yes

6. Review Comments to the Author

Reviewer #1: All the areas of my concern has been addressed by the authors. I do recommend for the publication of this manuscript.

Reviewer #2: "Overexpression of Medicago sativa LEA4-4 can improve the salt, drought, and oxidation resistance of transgenic Arabidopsis" was very interesting and easy to read, no doubt that the many corrections have greatly contributed to improve this work.

Reviewer #3: The reviewer is still not satisfied by the organellar marker (i.e. DAPI staining for nucleus) but the image is clear enough. The authors had answered the reviewer's questions and ready for acceptance.

7. PLOS authors have the option to publish the peer review history of their article (what does this mean?). If published, this will include your full peer review and any attached files.

Reviewer #1: Yes: Prof Fang Liu

Reviewer #2: No

Reviewer #3: No

---

## [Editor Report · Acceptance letter]

22 May 2020

PONE-D-19-31989R1 

Overexpression of *Medicago sativa LEA*4-4 can improve the salt, drought, and oxidation resistance of transgenic *Arabidopsis*

Dear Dr. Jia:

I am pleased to inform you that your manuscript has been deemed suitable for publication in PLOS ONE. Congratulations! Your manuscript is now with our production department. 

With kind regards,

on behalf of

Professor Keqiang Wu 

Academic Editor

PLOS ONE